# Transimpedance Matrix Measurement (TIM) Parameters Evaluation for the Assessment of Cochlear Implant Electrode Placement and Modiolar Proximity in Children

**DOI:** 10.3390/biomedicines13020319

**Published:** 2025-01-29

**Authors:** Katarzyna Radomska, Marcin Talar, Karolina Haber, Paulina Mierzwińska-Dolny, Andrew J. Fishman, Józef Mierzwiński

**Affiliations:** 1Department of Otolaryngology, Pomeranian University of Medicine, ul. Unii Lubelskiej 1, 71-252 Szczecin, Poland; 2Medicus Sp. z o.o., pl. Strzelecki 24, 50-224 Wrocław, Poland; 3Pediatric Cochlear Implant Program, Department of Otolaryngology, Audiology and Phoniatrics, Children’s Hospital of Bydgoszcz, Ul. Chodkiewicza 44, 85-667 Bydgoszcz, Poland; 4Department of Developmental Age Diseases, Nicolaus Copernicus University, 87-100 Torun, Poland; 5Department of ORL, Military Medical Academy, 11000 Belgrade, Serbia; 6Department of Otolaryngology, University of Missouri, Columbia, MO 65211, USA

**Keywords:** cochlear implant, objective measures, transimpedance matrix, child or children, inner ear malformations

## Abstract

**Introduction**: Transimpedance matrix measurement (TIM) is an electrophysiological measurement protocol of the impedance patterns of electrode contacts within the cochlea. Several studies have reported that TIM is an effective tool for the identification of abnormal electrode array placement. However, the normative values for properly inserted electrodes, as well as correlation of the TIM patterns with the electrode position, are not completely determined. **Objectives**: The first aim of this study is to establish normative values of TIM measurements obtained in children with proper electrode array insertion and tip fold-over, with proper inner ear anatomy and in congenital anomalies. The second aim of this study is to compare TIM measurements in Slim Modiolar (SM) and in Contour Advance (CA) electrodes, as their position is different according to the modiolus proximity. **Methods**: A total of 55 pediatric patients were included in the study and underwent cochlear implantation. 62 intraoperative measurements were conducted in this group—50 in children with normal inner ear anatomy and 12 in children with inner ear malformations. After each implantation, a plain x-ray was obtained. **Results**: There were clear statistically significant differences in TIM patterns in patients where electrode fold-over was confirmed and between SM and CA electrodes. **Conclusions:** TIM is a promising technique for intraoperative analysis of electrode placement. TIM patterns differ and correlate consistently with the different models of array implanted. This study is the first to report TIM patterns observed in children with normal inner ear anatomy and in inner ear malformations.

## 1. Introduction

In the present era, cochlear implantation has become a widely accessible method of rehabilitating patients with profound sensorineural hearing loss. The indications for treatment with this method are constantly evolving, and the indications are subject to change. This is due to the introduction of increasingly sophisticated technology in cochlear implants and the expansion of knowledge about electrophysiology, which has led to advances in surgical procedures. A cochlear implant (CI) is an electronic prosthesis for hearing that functions by capturing sounds from the surrounding environment and transforming them into electrical impulses, which stimulate first order auditory neurons by means of an intra-cochlear electrode array. The placement of the electrode in close proximity to the cochlear nerve allows for more precise stimulation due to the proximity to the nerve endings. Additionally, this configuration reduces energy consumption [1]. The application of a thin, delicate perimodiolar electrode, exemplified by the Cochlear 532/632 model, allows for a potentially close positioning to the modiolus [2], while simultaneously preserving residual hearing and facilitating localization within the scala tympani [2,3]. However, as demonstrated in other studies, the use of perimodiolar electrodes is more often associated with the risk of tip fold-over (TFO), which occurs when the electrode tip folds over the cochlear wall. The incidence of tip fold-over (TFO) for the Cochlear CI512/612 (Contour Advance—CA) electrode is 1.05%, while for the Cochlear CI532/632 (Slim Modiolar—SM) electrode, it may be as high as 10% [4]. An electrode bend within the cochlea may result in worsening of speech discrimination or facial nerve stimulation [5]. For this reason, great importance is attached to the possibility of intraoperative diagnosis of this complication and the appropriate intervention, namely electrode reinsertion. In order to achieve the aforementioned objective, it is essential to prioritize the potential for intraoperative diagnosis and the subsequent appropriate intervention, which may entail re-insertion of the electrode.

To assess the appropriate course of insertion and the placement of the electrode within the cochlea, one may utilize radiological examinations in the form of fluoroscopy [6], conventional radiography, or cone beam computed tomography [2]. The application of CT or RTG scans intraoperatively is associated with several disadvantages, including the need for space, equipment, and personnel who are not accustomed to working in surgical settings. Furthermore, radiological examinations expose patients, frequently children, and personnel present in the operating room to ionizing radiation. The performance of intraoperative imaging studies prolongs the duration of the surgical procedure. Consequently, in certain cochlear implant centers, these studies are not feasible intraoperatively and are instead conducted only after the first or second postoperative day. Such an approach precludes the possibility of modifying the electrode placement in the event of an anomaly, thereby exposing patients to the potential risk of reoperation.

One of the parameters that can be evaluated in radiological studies is the angle of bend of the electrode, which indirectly indicates the proximity of its position to the modiolus. This fact was used in the work of Perenyi [7], where the SM electrode was shown to be positioned closer to the modiolus than the CA. However, this test does not allow assessment of potential electrode dislocation to the scala vestibuli, which is important for the future functioning of the CI patient. In previous work, it was shown that translocation of the electrode to the scala vestibuli can lead to significant detriment in speech discrimination and requires greater consumption of the energy [2,7].

One of the technological limitations for totally implantable cochlear implants is the necessity of recharging the battery in the device. The idea is that a suitable electrode should be selected for this type of solution, and an intra-operative assessment of its position should be carried out. In order to provide the optimal conditions, electrophysiological measurements have been conducted, relying on the assessment of electrode impedance and electrically evoked responses of the auditory nerve. However, these techniques have proven inadequate for the intraoperative detection of TFO [5,8].

In their 2012 study, Vanpoucke and colleagues [9] proposed a method for measuring electric field strength based on the differences in passive voltage between adjacent electrodes in an implant. After dividing the voltage value by the stimulating current value, the impedance (ohms) value was obtained, which varies with the distance between the stimulating and recording electrodes [9,10]. Based on this, a 22 × 22 square matrix was constructed for the Cochlear electrode with 22 contact points, which depicts the transimpedance of the electrodes in the form of a heatmap [9,10,11].

When TIM measurements are obtained, the intracochlear potential is measured along each of the 22 electrode contacts while stimulating a single contact in monopolar mode. An example of this can be seen for a single electrode in Figure 1A and for all electrodes in Figure 1B. With a proper insertion the recording electrode is at its maximum value, in this case, at the corresponding electrode 9 diagrammed in Figure 1A. The voltage measured at the recording electrodes is divided by the applied current at the stimulation contact. Figure 1 shows TIM (A, B, C) results displayed in CustomSound^®^ EP software suite. When a single electrode is stimulated, it generates a voltage. The voltage is measured not only at the stimulation electrode itself, but also at all the other contacts of the electrode array. The farther the distance from the stimulated electrode, the lower the voltage. This measurement is repeated for each electrode of the array and all remaining electrodes. In a simplified way, this allows the assessment of the electrode position within the cochlea. Furthermore, it can be graphically displayed (Figure 1C) as an electrode position “image” called a heatmap, which can be easily assessed visually by the surgeons during the surgery.

The highest values (black boxes) are recorded at stimulating electrode and forms the diagonal with the bottom left corner of the matrix representing the apical (distal electrode 22) to the top right basal (proximal electrode 1). An open circuit is represented by light grey color and a short circuit by dark grey.

The evaluation of the proper positioning of the electrode in the context of a heatmap analysis is based on a subjective assessment of colors. However, as demonstrated by Leblans, Żarowski, and colleagues [11], in certain instances, the curvature of the electrode at its apex could lead to erroneous interpretations.

Figure 2A displays a TIM heatmap with a pattern typical for suspected tip fold-over. This heatmap suggest that electrodes 9 and 20 are close to each other (two light orange points). According to the legend, impedance values are represented from dark blue to red and should spread evenly from red to blue in the HeatMap. The increase in the impedance value caused by the proximity of, e.g., two electrodes (9 and 20) is visualized in this way on the heatmap. This color pack suggests a tip-fold over the electrodes in the cochlea. TFO was confirmed by CT scan, as seen in Figure 2B, where the array is kinked and folded upon itself in the region of the pars ascendans of the cochlear lumen.

Following further work on intraoperative measurements for cochlear electrodes [10,11], the Smart Nav application was introduced for the detection of intraoperative TFO based on transimpedance measurement. The aforementioned studies were conducted on individuals aged 18 and above.

Based on our research and analysis of published data, previous transimpedance electrode measurements have only been conducted on adults with normal cochlear anatomy. The presented study included children with both normal ear anatomy and those with congenital ear abnormalities. To the best of our knowledge, this is the first study to demonstrate the utility of transimpedance CI measurements in pediatric populations and in cases of anatomical anomalies.

The objective of this study was to analyze intraoperative transimpedance measurements of the cochlear implant electrode with the aim of detecting electrode bending and assessing the electrode position relative to the modiolus in pediatric patients.

Previously, measurements of transimpedance mapping (TIM) have been conducted using color assessment of the transimpedance map. The objective of the analysis presented in this study was to identify statistically measurable parameters that could be used to establish norms and detect the electrode bend, which can be used both in children with normal and with abnormal cochlea. The next objective was to determine the parameters that would enable the assessment of the position of the electrode in relation to the modiolus. To achieve this, the parameters for the Contour Advance and Slim Modiolar electrodes were compared, as the SM electrode is thinner and more delicate than the CA electrode and therefore positioned closer to the modiolus [7].

To our knowledge, the effectiveness of TIM measurements has not yet been evaluated in a pediatric group with normal cochlear anatomy and congenital anomalies. Furthermore, this is also the first attempt to apply TIM to the position of the CI electrode in relation to the cochlear modiolus.

## 2. Materials and Methods

The study was completed in accordance with the ethical standards of the institutional research committee and principles of the World Medical Association Declaration of Helsinki Ethical Principles for Medical Research involving human subjects and its later amendments. Ethical approval for this study was obtained from the local ethics committee.

We prospectively collected the data from three cochlear implant centers enrolled in the study. The inclusion criteria were as follows: age 18 years or less; cochlear implantation with intraoperative TIM measurement.

A total of 55 pediatric patients (32 females and 23 males) aged 1–17 years (mean age 4.6 years) were included in the study. A total of 62 measurements were conducted (38 in female ears and 24 in male ears). This included 50 with normal inner ear anatomy and 12 with inner ear malformations (enlarged vestibular aqueduct—3, incomplete partition II—3, incomplete partition III—3, cochlear hypoplasia—1, common cavity—1, cochlear aperture hypoplasia—1).

The approach to surgical intervention for congenital malformations of the inner ear entailed the utilization of a classical procedure via posterior tympanotomy. In instances where a round window was present, the round window approach was employed. In one case of common cavity defect, a “banana shape” approach was employed, utilizing a 24REST electrode [12]. In cases where there was a high risk of electrode insertion into the internal auditory canal, perimodiolar electrodes were utilized. In the case of EVA defects, Slim Modiolar (Cochlear, Sydnay, NSW, Australia) electrodes were utilized, while in other instances, Contour Advance (Cochlear, Sydnay, NSW, Australia) electrodes were employed. After each implantation, a modified Stenver’s plain X-ray was obtained intraoperatively.

### 2.1. Establishing Normative Values for the TIM Measurement

Traditional impedance and Auto-NRT measurements were first acquired as per protocol for all single electrode contacts. TIM was measured by standard means of Cochlear’s CustomSound^®^EP Suite tool using the Cochlear Nucleus CP910 processor and Cochlear Programming Pod. For each patient TIM measurements and heatmap of impedances was obtained. The TIM data was further analyzed looking for correlations, trends and dependencies between values.

The two following principles are considered when analyzing TIM measurements: monotonicity of waveform peaks and analysis of the diagonal of the matrix. The first principle assumes that the TIM measurement has exactly one peak in each recording. Based on this principle, there is also a single black maximal peak represented on the diagonal of the matrix (see Figure 1C). The pattern observed with a normal array in proper position should express a monotonic decrease in values moving away from this single peak in either direction. (see Figure 1B). The principle also assumes the diagonal of the matrix is derived from the largest impedance values according, as was described by Vanpoucke [9]. When moving away from the diagonal, the values should decrease on the heatmap (see Figure 1C). Additional peaks are therefore interpreted as occurrences of non-monotonicity.

Statistical analysis of obtained values was based on 6 parameters of TIM measurement:

1. Number of points of non-monotonicity, defined as points in the matrix, that are exempted from the principle of monotonicity. The optimal number of these points in each line should be 0. The function calculating these points allows to determine the sensitivity, i.e., whether each trend violation will be taken into account, or only a violation by a certain percentage of the median.

2. Number of peaks in the matrix. The optimal number of these points in each line should be 1. The function calculating these points allows to determine the sensitivity, i.e., whether each local extreme will be taken into account, or only one that is greater by an appropriate percentage than the bordering points.

Figure 3A shows voltages measured at one electrode. Normally, there is one distinct peak for a stimulated electrode. Figure 3B shows the abnormal cycle of inducing voltage at more than one electrode. Two peaks are visible. One primary peak is at electrode E10, and a second peak representing an occurrence of non-monotonicity is at electrode E17. When the expected values change, which should decrease and instead increase, we mark non-monotonic points. A detailed analysis of the heatmap values reveals the presence of peaks that may indicate potential obstacles in the path of the impulse during impedance measurement. However, in some instances, these values may be so minimal that they are not discernible in the heatmap, becoming visible only during subsequent analysis.

To assess the whole 22 lines of TIM, the following 4 parameters were assessed:Mean number of points of non-monotonicity;Mean number of peaks in the matrix;Maximum number of points of non-monotonicity;Maximum number of peaks in the matrix.

To assess the electrode position within the cochlea, the following two parameters were assessed:5.Correct construction of the diagonal;6.Minimalities on the sub-diagonals.

These 2 parameters are based on determining 22 maximum impedance values (black line) and 42 maximum transimpedance (green lines) values after excluding the diagonal. The first measure is to determine how many of the values are on the diagonal; the best situation is when these are all 22 values. This value results from the number of electrodes in the cochlear implant. The second one describes how many of the values are on the two sub-diagonals adjacent to the main diagonal; optimally, these should be all 42 values in the matrix. The optimal position of the electrodes in the modiolar CI electrode will have 42 points on the sub-diagonals. Figure 4 shows a matrix diagonal and 2 sub-diagonals that lay 1 column below and 1 above the diagonal.

For all above mentioned parameters, a comparison was conducted between the values of the following transimpedance parameters in a group of patients with proper electrode placement in the cochlea and the values obtained in a group of patients with TFO and to compare electrodes CA and SM.

### 2.2. Statistical Analysis

The sensitivity of the function for calculating non-monotonicity points was assumed in the analysis at the level of 1% of the row median; i.e., only such violations of the expected trend are taken into account, which differ by at least 1% of the median compared to the previous value. The sensitivity of the function for calculating peaks was assumed at the level of 1.02; that is, only values that are greater than 1.02 times the values bordering them are taken into account as local extremes. Mean and 95% confidence intervals of the mean were taken for all parameters. Statistical analyses were conducted using R version 4.1.0 (R Core Team, 2021) in IDE RStudio version 1.4.1717 (RStudio Team, 2021). Visualizations were made using R version 4.1.0 and JASP (version 0.15.0.1).

## 3. Results

A total of 62 measurements were conducted in a group of 55 children. This included 50 measurements with normal inner ear anatomy and 12 with inner ear malformations. Electrode tip fold-over

TIM measurements indicated an electrode array position without TFO in 57 out of 62 patients (92%). Five tip fold-overs were observed intraoperatively (8% of all cases). One tip fold-over was diagnosed in a case with a malformed inner ear (8% of this group), and four instances in cases with a normally formed inner ear (8% of this group). Interestingly, electrode tip fold-overs occurred only when the CI632 electrode was implanted. The examples of modified Stanvers x-ray is shown on Figure 5.

In implants where fold-over was diagnosed, the number of points of non-monotonicity and number of peaks in the matrix were statistically significantly increased. The highest value for mean number of points of non-monotonicity in the fold-over group (*n* = 4) was 2.091. The highest value in the group with proper electrode position (*n* = 42) was 1.55. In 42 out of 46 measurements performed in this group mean number of points of non-monotonicity was equal to 0. Other parameters, including the construction of the diagonal and minimalities on the sub-diagonals did not reach statistical significance. All data are presented in Table 1 and Appendix A. Mean and 95% confidence intervals of the mean for peaks are presented in Figure 6. 

### Comparison of Implant Electrode Array Models

Calculations were made using the normal inner ear implantations, excluding cases with electrode tip fold-over, and were based on a total of 46 measurements.

The sole parameter with statistical significance was minimalities on the sub-diagonals. This parameter can differentiate the type of electrode between Cochlear™ Nucleus^®^ CA and SM. All other comparisons were not statistically significant. Detailed data are shown in Table 2. 

## 4. Discussion

The use of cochlear implants as hearing prostheses is highly efficacious. In numerous studies, it has been demonstrated that appropriate surgical techniques and electrode placement in the cochlear scala tympani are crucial for subsequent rehabilitation and speech discrimination [13,14]. It is therefore essential to develop intraoperative techniques that allow for the assessment of electrode placement accuracy within the cochlea. The use of perimodiolar electrodes presents an additional surgical challenge due to the potential risk of TFO. As previously stated, the use of radiological imaging in surgical settings presents significant challenges, with some institutions even reporting its unfeasibility. The utilization of electrophysiological measurements appears to be the optimal solution, as their implementation only marginally extends the duration of the procedure, while the requisite hardware concerns primarily a computer with the necessary software.

### 4.1. Electrode Tip Fold-Over (TFO)

The phenomenon of the electrode implant electrode tip bending is widely discussed in the literature, but there is no consensus among authors on the appropriate course of action in such cases. Published reports indicate that the detection of TFO in CI patient groups necessitates the deactivation of certain electrodes [5] or even that no action be performed [5]. Our analysis of the data indicates that the optimal placement of the electrode in the tympanic scala allows for the most effective rehabilitation outcomes. This is particularly relevant for pediatric patients, given that the assumption is that the implanted device should perform its function throughout the patient’s lifetime. For this reason, we believe that the detection of TFO during surgery and reinsertion of the electrode is essential. The ease of detecting this complication allows for the use of electrophysiological studies. The detection of TFO based on electrophysiological measurements employs a variety of measurement techniques. As detailed in the research conducted by Verghese and colleagues, [15] the authors proposed electrocochleography, which demonstrated excellent sensitivity. However, specificity varied depending on the applied parameters. The success rate ranged from 41% to 68.9%, which is not an acceptable outcome when one considers the necessity of making decisions regarding electrode re-insertion and the potential trauma to the delicate cochlear structures. However, the electrocochleography method is considered effective for assessing residual hearing, both in manual electrode insertion [16] and through the robotic surgery [17]. Simultaneously, studies were conducted on the utilization of electrically induced potentials from the auditory nerve (eCAP). However, as demonstrated in the Grolman [8] and Zuniga [5] studies, eCAP-based electrophysiological measurements were not highly effective in detecting TFO and required significant experience from the audiologist evaluating the test. Therefore, they did not meet the requirements regarding sensitivity and specificity of the method.

Proper insertion of the electrode array is crucial to outcome, and this application of TIM is most widely described in the literature. The prevalence of fold-overs for all types of electrode arrays in a meta-analysis is reported to be 2% [18]. This is best addressed with immediate repositioning; therefore, early detection of this problem is crucial. Despite low overall tip fold-over rates, the problem can be more prevalent in perimodiolar electrodes [4,10,18]. Due to the precurved design and insertion technique, these seem more susceptible to fold-over [4,10]. For CI632 Slim Modiolar electrodes used in other studies, tip fold-over rates vary between 4.3 and 10.5% for normal cochleas [19]. Various predisposing factors for electrode fold-over have been assessed in prior studies. For electrodes implanted in our study, slight modifications in insertion techniques that can sometimes go unnoticed may have caused this complication, especially in difficult implantations. Some reported concerns include the white handle surgical sheath being misaligned, deep insertion of the sheath in the cochlea, and electrode insertion with the tip of the array starting out of the sheath [20]. With the growing trend towards residual hearing preservation and construction of thinner and more delicate electrodes, the risk may even be higher in the future. When tip fold-over occurs, the voltages at the electrodes lying close to each other are similar. Also, the highest maximum value should still be at the stimulating electrode, but another local maximum will appear. The current version of the Smart Nav algorithm and the measurements proposed in this study allow for the detection of additional local peaks. In our study, we propose the detection of TFO in a measurable format, which may prove beneficial in subsequent investigations related to the utilization of this type of measurement, particularly in the context of robotic-assisted surgery and artificial intelligence algorithms. Electrode tip fold-over can involve larger and shorter parts of the electrode. It can involve only an electrode at the tip with few contacts, or the array can fold in the middle [11,20]. Electrode malinsertion cannot always be felt by the surgeon and may be difficult to identify at the time of surgery [10]. It can go undetected until later problems emerge intra- or post-operatively, resulting in possible facial nerve stimulation or poor performance [10]. Therefore, intraoperative diagnosis of such complications is of particular importance in pediatric patients. In our study, we demonstrated for the first time the efficacy of measurements based on TIM in children with both normal and abnormal anatomy of the internal ear. It is suggested that parameters proposed in this paper may be used for all types of electrodes from different manufacturers, but this requires further investigation.

### 4.2. Perimodiolar Electrode Position and Intraoperative Measurements

The current version of the Smart Nav application has not impeded the advancement of research into the potential of TIM for assessing the positioning of electrodes close to the modiolus. This is due in part to the fact that it allows for the assessment of potential electrode bending but does not provide the capability to evaluate all transimpedance parameters.

Salkim et al. [21] conducted experimental studies on cochlear models, measuring impedance with simultaneous electrode position assessment. They demonstrated the intraoperative utility of these measurements in determining the position of the electrode relative to the scala tympani border, and their study confirms the need for further research into the possible use of CI electrode impedance measurements.

Another study indicating the usefulness of impedance measurements in the evaluation of electrode placement is that of Gottfried et al. [22]. In their study [22], the impedance measurements performed during electrode insertion, in conjunction with the eCAP readings, allowed the monitoring of electrode placement. Furthermore, given the optimal perimodiolar arrangement of the electrode, it would be prudent to consider the results published by Aschendorff and colleagues [2]. The aforementioned researchers demonstrated that in instances of excessive insertion depth, called “over-insertion” of the electrode SM, there is a tendency for the electrode to deviate from the helical pitch at the bend in the shaft. With this problem in mind, Lee and colleagues [23], in their experimental study, employed the pull-back technique, which yielded more favorable results in eCAP parameters [23]. This maneuver was implemented with the objective of preventing over-insertion. Recently, measurements have been conducted based on impedance measurements in Riojas’s studies [24,25,26]. These measurements were taken during robotic insertion and demonstrated the utility of impedance measurements in the vicinity of the electrode in relation to the cochlear modiolus. In view of the findings of earlier research, it appears rational to initiate additional studies on the utilization of the parameters suggested in our own research to differentiate the SM electrode from the CA electrode as a means of evaluating the position of the electrode in relation to the cochlear spindle. Nevertheless, this endeavor necessitates the execution of studies on a more substantial cohort of patients.

The introduction of the Smart Nav application undoubtedly facilitated daily work on the surgical theater and enabled intraoperative detection of TFO. However, the requirements imposed on implantable hearing devices are considerable. Such devices are already implanted in children below the age of one, with the expectation that they will be used throughout their lifetime. Electrophysiological measurements of cochlear implants retain considerable potential for application in both conventional insertions and the rapidly evolving field of robotic surgery [17]. Nevertheless, further research is required to determine which parameters will be applicable in clinical practice and to identify suitable, reproducible parameters for this purpose.

It would be optimal to create a tool based on different parameters. Such a tool would enable surgeons to determine the optimal position and depth of insertion for cochlear nerve stimulation intraoperatively, while preserving residual hearing. The application of an algorithm that employs both eCAP and transimpedance parameters would facilitate the optimal positioning of the electrode during surgical procedures, both those performed manually by the surgeon and those conducted using robotic surgery. Such electrode positioning will facilitate superior rehabilitation outcomes while simultaneously reducing future energy consumption by the fully implantable CI. As is widely acknowledged, the objective is to develop fully insertable implants. However, research is still being conducted to identify methods of obtaining the requisite electrical energy for the optimal functioning of the implanted hearing system. In order to achieve the aforementioned objectives, it is necessary to have tools that allow for a statistically measurable evaluation of results. One such tool is the proposed parameters describing the possibility of TFO (mean number of points of non-monotonicity and number of peaks in the matrix), as well as the mean number of points of non-monotonicity, which can indicate indirectly modiolus proximity.

## 5. Conclusions

The small size of the study group, especially of cases where TFOs occurred, does not allow categorical conclusions to be drawn. However, it appears that TIM measurement can be considered as reliable method for detecting electrode array tip fold-over intraoperatively. The measurement can be conducted with no additional radiation exposure, which is especially important in children. It can be used as a cross-check method and even has the potential to fully replace intraoperative x-ray-based imaging. Our preliminary results confirm that TIM measurement with our methodology diagnoses TFO in all our cases, but the establishment of normative values for the proposed parameters requires further research on a larger group of patients in order to use this tool as a reliable method for assessing the position of the electrode and its relation to the modiolus.

To our knowledge, this study is the first to report experience with TIM in children with normal cochlear anatomy and inner ear malformations. We believe that in inner ear malformations, TIM measurement is a quick, intraoperative measurement that may confirm good electrode position within the cochlea in difficult cases.

Our study complements the knowledge about the use of research based on transimpedance matrix measurement and opens completely new possibilities for using this tool on large databases, e.g., in the context of using AI-based tools and robotic surgery.

## Figures and Tables

**Figure 1 biomedicines-13-00319-f001:**
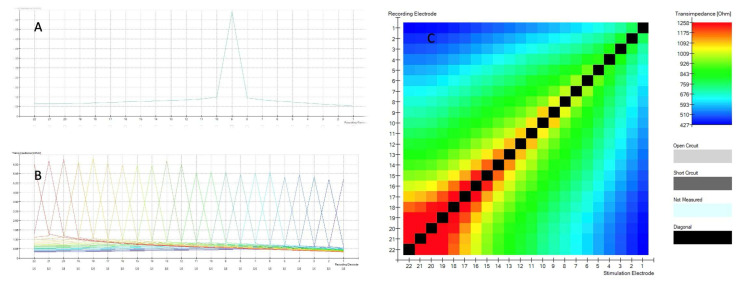
(**A**) Voltage of electrode 9. (**B**) Voltage of electrode 22. On the horizontal axis, the numbers of individual electrodes are marked; on the vertical axis, the measured voltage. (**C**) Heatmap.

**Figure 2 biomedicines-13-00319-f002:**
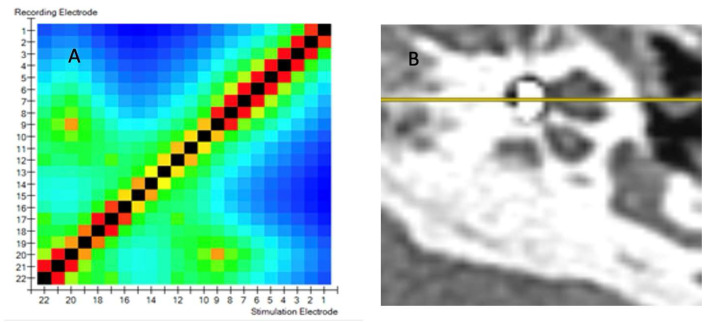
(**A**) TIM heatmap with a pattern typical for suspected tip fold-over. (**B**) TFO confirmed by CT scan.

**Figure 3 biomedicines-13-00319-f003:**
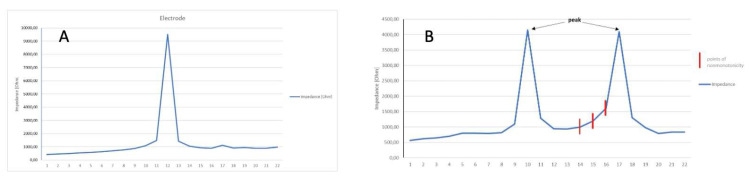
(**A**) Voltage measured at one electrode. (**B**) Abnormal cycle of inducing voltage at more than one electrode.

**Figure 4 biomedicines-13-00319-f004:**
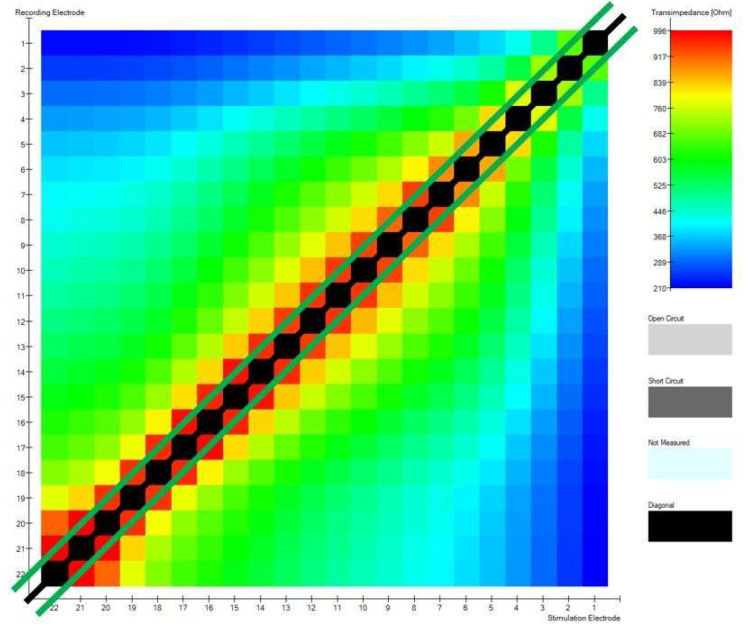
TIM heatmap with the matrix diagonal as a black line from the bottom left corner of the matrix (apical) to the top right (basal). There are, accordingly, two sub-diagonals, which are presented as green lines in the figure.

**Figure 5 biomedicines-13-00319-f005:**
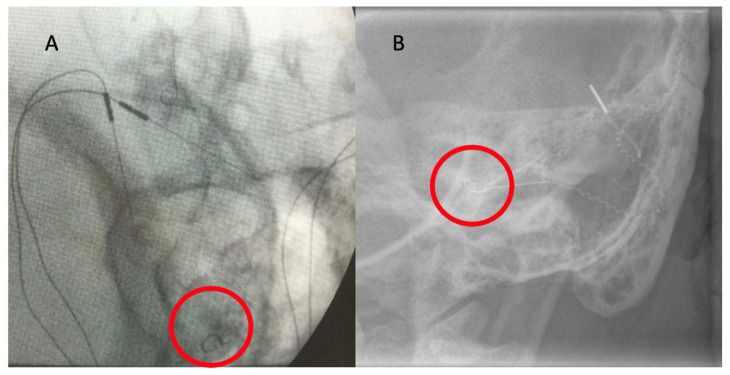
Modified Stanvers plain radiography performed intraoperatively. Tip of the electrode is marked with red circle. (**A**) Normal electrode tip position. (**B**) Tip fold-over (TFO).

**Figure 6 biomedicines-13-00319-f006:**
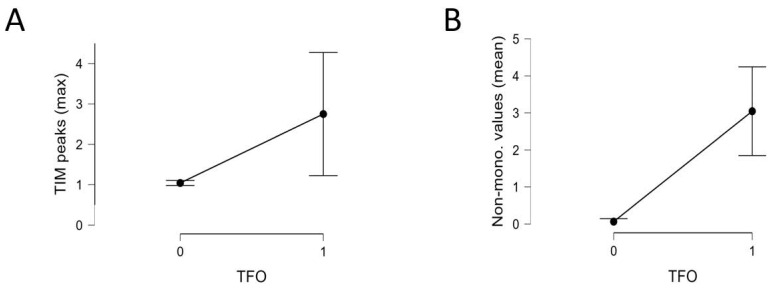
(**A**) Mean and 95% confidence intervals of the mean for peaks (TIM peaks (max) = maximum number of peaks) with (0) and without (1) TFO groups. (**B**) Mean and 95% confidence intervals of the mean for non-monotonic values with (0) and without (1) TFO groups.

**Table 1 biomedicines-13-00319-t001:** Results of Student’s *t*-test in all cases with detected fold-over vs in a group with proper insertion.

Measure	Mean Value—in Fold-Over Cases	Mean Value—Proper Insertion Cases	*T* Test	df	*p*
mean number of points of nonmonotonicity	3.045	0.064	−7.882	3.066	0.004
2. mean number of peaks in the matrix	1.727	1.041	−4.877	3.272	0.013
3. max. points of nonmonotonicity	5.250	0.109	−10.667	3.082	0.002
4. maximum number of peaks in the matrix	2.750	1.043	−3.558	3.024	0.037
5. correctness of the diagonal construction	0.750	0.935	0.731	3.131	0.516
6. minimality on sub-diagonals	32.250	31.783	−0.107	3.124	0.921

**Table 2 biomedicines-13-00319-t002:** Comparison of Cochlear™ CA and SM electrode arrays.

Measure	Mean Value in CA Electrode	Mean Value in SM Electrode	*T* Test	df	*p*-Value
mean number of points of non-monotonicity	0	0.098	−1.655	29	0.109
2. mean number of peaks in the matrix	1	1.063	−1.437	29	0.161
3. max number of points of non-monotonicity	0	0.167	−1.980	29	0.057
4. maximum number of peaks in the matrix	1	1.067	−1.439	29	0.161
5. correctness of the diagonal construction	1	0.900	1.795	29	0.083
6. Minimality on sub-diagonals	29.750	32.867	−2.258	29	0.034

## Data Availability

The raw data supporting the conclusions of this article will be made available by the authors on request.

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
