# Peer review of "Transimpedance Matrix Measurement (TIM) Parameters Evaluation for the Assessment of Cochlear Implant Electrode Placement and Modiolar Proximity in Children"

_biomedicines, 2025, doi:10.3390/biomedicines13020319_

Round 1

Reviewer 1 Report

Comments and Suggestions for Authors

Introduction:

- The introduction sets the scene well and explains why TIM was a suitable choice for the measurement of cochlear implant electrode placement and modiolar proximity in children.

- It underlines the need for accurate placement of electrodes to optimize implant function and the limitations of current intraoperative imaging techniques.

- To improve the introduction, include more specific information on why TIM is better than previous methods and which gaps in literature are fulfilled by this study. cite doi:10.3390/audiolres12040042.

- Clearly stated and focused with the major goals of the study to determine normal TIM values in children with normal and abnormal inner ear anatomy and to compare TIM between different electrode array models.

Methods:

- Study has a well-structured methods section which thoroughly describes study design, patient population, measurement protocols, and statistical analyses.

- Go into a little more detail regarding surgical approach and electrode array selection for the various inner ear anomaly groups.

- Explain how you differentiate "proper electrode placement" from "tip fold-over" according to the observed radiology findings.

- Justify the sensitivity thresholds chosen in non-monotonicity and peak detection functions.

Results:

- The Results section should give you the key findings in a concise format, along with helpful information about the statistical analyses that were performed.

- You might want to add a table or figure summarizing the TIM parameter values across the different groups (normal vs. abnormal anatomy, CA vs. SM electrodes).

- Please describe the specific TIM patterns, in addition to the statistical differences observed in the measured parameters, in the two cases with tip fold over.

Discussion:

- The discussion appropriately puts the study's findings in context and emphasizes the clinical relevance of the results as well as the novelty of evaluating TIM in a pediatric group with a spectrum of inner ear anomalies.

- Elaborate on the possible mechanisms contributing to the TIM differences identified between the two electrode array models and their positioning in relation to the modiolus. cite doi: 10.1016/j.otc.2019.09.004

- Address the limitations of the investigation, including the small sample size of the inner ear anomaly subgroup and potential sources of variability in TIM measurements.

- Suggest practical clinical applications of the reported TIM normative values and parameters to influence intraoperative decision-making and enhance cochlear implant outcomes.

Conclusion:

- Conclusion Summarizes the key findings of the study and highlights the potential of TIM as a novel tool that may prove useful for intraoperative optimization of cochlear implant electrode positioning.

Author Response

Dear Reviewer

Thank you very much for all your comments on the presented research results. I hope that the revised version of the manuscript will meet expectations and can be published in the journal Biomedicines. All responses are described in the attached file.

Reviewer 2 Report

Comments and Suggestions for Authors

This manuscript summarizes a study of a relatively new technique to provide a comprehensive analysis of impedances across a cochlear implant electrode array during surgical placement.  This topic is of great interest in the field due to the significant occurrence of electrode array tip fold-overs and other misplacements of electrodes during surgery.  Although radiological imaging during surgery can identify such problems, this is not feasible in all locations.  The technique (Transimpedance Matrix Measurement or TIM) may provide suitable feedback to surgeons to allow re-insertion or replacement at the time of initial surgery.  Electrode issues have been shown to have possible detrimental effects on hearing outcomes for patients.  This preliminary investigation of the technique looks at a moderate sample of pediatric cases and reviews a number of electrophysiological parameters that may be useful in identifying tip fold-overs in particular.  There is also a comparison of two different electrode designs that may suggest methods for determining additional aspects of electrode placement (eg. proximity to the modiolus).  I have identified some significant problems with the writing (identified on a pdf file attached to this review) but I think that the data and its analysis is of interest.  One significant issue is that there is a main conclusion that TIM measurement is "a quick and reliable method for detecting electrode tip fold -over intraoperatively".  I do not feel that the evidence presented is adequate to support this strong conclusion.  There are significant differences for a number of measurements on a group basis (Table 1 shows these differences between fold-over cases and normal insertions).  Figure 5 goes further showing a plot of results for two parameters for the fold-over cases (note that there are only 5) and some of the normal insertions (I count 12).  The reader needs to see all the normal cases plotted on this chart to make a judgment about whether the main conclusion is supported. The group differences are encouraging, but for the TIM to become a reliable clinical tool it needs to have both high sensitivity and specificity.  A plot of all the data as in figure 5 would demonstrate how well the technique can perform.  Even with this analysis, there probably needs to be a larger sample to be sure of the efficacy.  I have identified a number of confusing sections of the text that I believe will need some rewriting (see attached file), and I feel there is a need for a better explanation of exactly how this technique works. One issue is why are the transimpedances higher around the active electrode.  A larger voltage peak would suggest a lower impedance based on simple Ohm's Law calculations.  Please see attached pdf file for detailed comments.

Author Response

Dear Reviewer,

We are immensely grateful for your comments, which have considerably enhanced the quality of the research results presented. Minor corrections have been made, while comments requiring more extensive responses have been added to the submitted PDF file and below. On behalf of all co-authors, it is hoped that the submitted responses are satisfactory and will allow the paper to be published in the journal Biomedicines. All responces are described in the attached file. 

With best regards

Katarzyna Radomska

Round 2

Reviewer 2 Report

Comments and Suggestions for Authors

Thank you for addressing the issues I raised in reviewing the manuscript.  I hope that the TIM method will prove to be helpful in optimizing cochlear implant electrode placement in the future, As with many measurements in this field it will probably need to be interpreted carefully along with other information to provide the best chance of success.